# Recent Advances in the Etiology, Diagnosis, and Treatment of Marmoset Wasting Syndrome

**DOI:** 10.3390/vetsci12030203

**Published:** 2025-02-27

**Authors:** Jaco Bakker, Remco A. Nederlof, Job Stumpel, Melissa A. de la Garza

**Affiliations:** 1Animal Science Department, Biomedical Primate Research Centre, 2288 GJ Rijswijk, The Netherlands; 2Independent Researcher, 2861 XZ Bergambacht, The Netherlands; remco.a.nederlof@gmail.com; 3WILDLANDS Adventure Zoo Emmen, 7801 BA Emmen, The Netherlands; j.stumpel@wildlands.nl; 4Michale E. Keeling Center for Comparative Medicine and Research, University of Texas MD Anderson Cancer Center, Bastrop, TX 78602, USA; made4@mdanderson.org

**Keywords:** marmoset wasting syndrome, weight loss, alopecia, diarrhea, hypoproteinemia, callitrichids, muscle atrophy, marmosets, microbiome, probiotics

## Abstract

Gastrointestinal diseases are a frequently encountered problem among captive callitrichid colonies. One of the most challenging health issues is Marmoset Wasting Syndrome (MWS). MWS is characterized by progressive weight loss, muscle atrophy, alopecia, anemia, hypoproteinemia, and chronic diarrhea. Several etiologies have been proposed for MWS, including stress, malabsorption, infection, and improper nutrition. Nevertheless, the exact pathogenesis remains unknown. Even though recent advances in treatment have been made, the prognosis remains poor.

## 1. Introduction

Callitrichids are small arboreal neotropical nonhuman primates (NHPs) that can often be found in both zoological parks and laboratories throughout the world. Of the callitrichid species, the common marmoset (*Callithrix jacchus*) is the species most often encountered in research colonies. Globally, the understanding and overall quality of veterinary care, nutrition, housing, husbandry, and handling practices have increased over the past decades [1]. Improvements in animal welfare have been achieved by reducing stress, offering species-specific diets, and providing opportunities to engage in species-specific behaviors.

Callitrichids are predisposed to several spontaneous diseases (e.g., gastrointestinal (GI) disorders) and related health issues that might negatively affect their well-being and reduce survival rates [2,3,4,5,6,7,8]. By using modern husbandry and handling practices, many of these diseases and conditions are well-controlled and present minimal risk to captive colonies. Nonetheless, the most devastating GI disease in captive callitrichids is an Inflammatory Bowel Disease (IBD)-like condition, commonly known as Marmoset Wasting Syndrome (MWS). Interestingly, MWS has never been observed in wild individuals, and is considered to affect only captive animals. It is most described in common marmosets but is also reported in other callitrichid species, such as other marmosets (*Callithrix* spp.), tamarins (*Saguinus* spp.), lion tamarins (*Leontopithecus* spp.), and pygmy marmosets (*Cebuella* spp.) to a lesser extent [9,10,11,12]. MWS encompasses a variety of disorders that are not well understood and lack an identifiable cause, hence the designation syndrome rather than disease.

The following is a comprehensive review of recent advances concerning the etiology, diagnosis, and treatment of MWS. These advances should be considered by those working with and caring for these animals, so that optimal health, welfare, and research will be achieved. For this review, a literature search for books, book chapters, peer-reviewed publications, conference proceedings, and newsletters in academic literature databases such as PubMed, Scopus, Web of Science, and Google Scholar was performed. We used search terms such as marmosets, callitrichids, Marmoset Wasting Syndrome, chronic weight loss, and stress to identify potentially relevant publications. We then evaluated reports that we considered as clinically relevant. Our review revealed large gaps in our overall understanding of MWS and its causes, prevention, and treatment options. These gaps may be used to direct future MWS research.

## 2. Prevalence

In captive marmosets, MWS is one of the most reported and widespread afflictions [3,4,5,6,9,13,14,15,16]. One study reported that up to 60% of the 86 euthanized captive laboratory-housed marmosets had evidence of MWS [5]. Another study stated that 31–44% of the 150 studied necropsy reports involved GI diseases, which may suggest MWS involvement. Unfortunately, the authors did not describe the causes of GI disease [17]. Other authors reported that idiopathic GI diseases (i.e., MWS, chronic malabsorption, and chronic diarrhea) were present in 1–18% of surveyed laboratories, involving approximately 2000 marmosets [2]. The prevalence of MWS across the European Association of Zoos and Aquaria (EAZA) callitrichid population (*n* = 1218 individual callitrichids) was assessed in 2017 and determined to be 13.1% [16]. In the literature, no sex or age predisposition for MWS is described. However, the reported prevalences may be inaccurate, given that one of the primary challenges surrounding MWS is that the term lacks diagnostic specificity.

## 3. Clinical Signs and Pathology

MWS is characterized by a generalized deterioration of body condition. Affected marmosets are reported to have a variety of clinical signs, including skeletal muscle atrophy, marked progressive weight loss, chronic or intermittent diarrhea, and alopecia (especially at the tail) (Figure 1), among other signs, or may result in acute death [4,5,9,13,14,15,18,19]. Associated clinical signs also include weakness and paralysis of the hind legs, wet or greasy fur, and a failure to thrive. MWS may present with concurrent musculoskeletal disease that may cause pain or result in pathologic fractures [20].

Blood tests have revealed various abnormalities including anemia, hypoproteinemia, hypoalbuminemia, and increased levels of the liver enzymes serum aspartate aminotransferase (AST) and alkaline phosphatase (ALP) [4,5,9,13,14,15,18,19].

Histopathological examination has associated the occurrence of chronic colitis, gastritis, hemosiderosis, nephritis, pancreatitis, hepatic lesions, parasitism, and segmental lymphoplasmacytic enteritis with MWS [4,5,9,13,14,21]. Histology reveals chronic lymphoplasmacytic inflammation in the intestines. Primarily lymphocytes are observed in the lamina propria and epithelium with villous atrophy, fusing and blunting, and crypt hyperplasia [4,5,9,13,22].

Phenotypic characterization of the intestinal inflammation should be completed using immunohistochemical techniques, which reveals a T-cell-mediated local immune response [22,23].

## 4. Etiology, Pathogenesis, and Predisposing Factors

While several hypotheses have been proposed, including inadequate husbandry practices, nutritional imbalances, vitamin deficiencies, alterations in the intestinal microflora, excessive stress, food sensitivity (i.e., gluten), and parasitic infestation, the exact etiology and underlying mechanisms of MWS have not been delineated and are likely multifactorial [1,4,14,16,23,24,25,26,27,28,29,30,31]. A recent MWS study, involving four healthy and four MWS-affected marmosets, indicated that inflammatory intestinal changes play a decisive role in the pathogenesis of the syndrome [22]. The authors suggested that cell proliferation continues, but cell differentiation is halted in the intestinal tract due to the upregulated β-catenin/Tcf7L2/Cyclin D1 signaling pathway in MWS, which results in mucosal dysfunction and subsequent villous atrophy.

### 4.1. Nutrition

Most research concerning MWS has focused on the dietary aspect of MWS. MWS has been assumed to be caused by nutritional deficiencies or imbalances, likely concurrent with varying degrees of nutrient malabsorption [24,28]. Wild marmosets eat insects, fruits, and gouge bark to obtain tree gum, which consists of a mix of polysaccharides, proteins, minerals, and water [1,32,33,34]. Wild marmosets typically spend approximately 70% of their time foraging tree gum, and up to 40% of their diet may consist of gum. The fiber in gum may promote the growth of healthy gut bacteria. However, in captivity, commercially available callitrichid diets are offered as complete food, supplemented with a variety of fruits and animal proteins like boiled eggs and dairy products [2]. This diet has resulted in several health issues, likely including MWS, as it provides the required nutrients but may not be ideal for their natural feeding behaviors or a diversified gut microbiome. These diets have gradually been changed to ones that are more naturalistic, consisting of insects, gum Arabic, and a limited amount of fruit and vegetables, or even no fruits and vegetables at all [33,34]. The dietary fiber in gum provides support to healthy gut bacteria, which are assumed to play a crucial role during gastrointestinal inflammation. A lack of these beneficial bacteria in the gut may lead to uncontrolled intestinal inflammation, as is the case in MWS.

Several causes and triggers for GI inflammation have been suggested and identified, such as protein deficiency, the presence of dietary gluten, low dietary fiber, low vitamin E concentrations, and GI dysbiosis. This inflammation may lead to chronic malabsorption and the manifestation of clinical signs associated with MWS [4,5,9,16,24,25,31,35,36,37]. A more recent study described that α1-proteinase inhibitor (α1-P1) concentrations in feces were increased, reflecting intestinal protein loss in animals with chronic lymphocytic enteritis and chronic weight loss [38]. Risk and protective factors related to dietary components and husbandry practices associated with MWS were identified in a survey across the EAZA community [16]. Surprisingly, crude protein levels were revealed not to be a significant predictor for developing MWS. In addition, the discrepancy between captive diets and wild diets was noted. The gum tree has a crude protein percentage significantly lower than that found in captive callitrichid diets (4–7% vs. 25–30%, respectively) [16,39]. Furthermore, it was suggested that the dietary crude protein intake by wild marmosets will never reach 25%, indicating a low probability that a relationship between protein deficiency and MWS exists.

This constitutes a contradiction to the Cabana et al. (2018) study on the role, or lack thereof, of protein deficiency in MWS [16]. Although an increased crude protein intake does not appear to act as a protective factor against MWS [16], protein supplementation in MWS-affected marmosets might alleviate signs by replacing GI protein losses [38].

Gluten is expected to be involved in MWS [25,40]. Gliadin is one of the main proteins in gluten. The levels of IgA-gliadin antibodies (IgA-AGAs) and IgA-containing circulating immune complexes (IgA-CICs) and the degree of glomerular IgA deposits between marmosets exhibiting clinical signs of MWS (*n* = 11) and animals not affected by MWS (*n* = 46) were compared [40]. Increased IgA-AGA and IgA-CIC levels in marmosets with MWS were observed. There was a significant correlation between the glomerular IgA deposition and the titer of IgA-AGAs. The group of marmosets strongly positive for glomerular IgA deposits comprised significantly more animals with MWS than the group without deposits. The results of this study suggest that antibodies against food-derived gliadins may be involved in the etiology and pathogenesis of MWS.

### 4.2. Gastrointestinal Flora

Changes in the fecal microflora of MWS-affected marmosets included an increase in the number of putrefactive organisms (*Bacteroides*-*fusobacteria*) and *Clostridium* sp., as well as a significant decrease in the numbers of lactobacilli. Populations of non-lactose fermenting organisms such as *Proteus* sp., *Pseudomonas aeruginosa*, and *Alcaligenes faecalis* were also reported to increase [4,34,41]. In pied tamarins (*Saguinus bicolor*), an increase in *Helicobacter jaachi* and *Lactobacillus* sp., as well as a relative decrease in *Lachnospiraceae* and *Ruminococcaceae*, was observed in affected animals [12]. Prior research did not find a difference in the abundance of *Helicobacter* sp. between affected and unaffected animals [42,43], but no analysis at the species level has been performed, and it is postulated that *Helicobacter saguini* and *Helicobacter jaachi* may behave differently within the gastrointestinal tract (GIT) [12]. *Lachnospiraceae* and *Ruminococcaceae*, which were present in fewer numbers in affected tamarins, have been demonstrated to have an anti-inflammatory effect in the GIT [44,45]. Both stress and low dietary fiber may contribute to a decreased presence of *Faecalibacterium*, an important species in the *Ruminococcaceae* family [46,47,48].

### 4.3. Stress and Housing Conditions

Stress has also been proposed as a predisposing factor for MWS, as it is described to suppress protective immune responses [16,49,50]. However, some individuals are perhaps more stress-tolerant than others, which makes establishing a direct correlation between stress and MWS challenging. In addition, the variation between captive environments makes it difficult to pinpoint exact triggers. Currently, it is even not exactly known whether chronic stress initiates MWS or merely worsens signs in subclinical individuals. Assessing the cortisol concentration in hair could be a useful tool, as quantifying cortisol concentrations in hair is known to be a noninvasive biomarker of long-term hypothalamic–pituitary–adrenal activation in callitrichids and thus can provide important information on its possible role in MWS [51].

In laboratory facilities and zoos, cage and habitat designs are often not naturalistic and provide insufficient sleep areas. Hiding places are often absent, eliminating the possibility for the animal to retreat from perceived threats. Animals may be confined to relatively small cages with no access to sunlight. In captive conditions, animals are often unable to engage in natural behaviors, e.g., tree gouging, hole making, and scent marking. Moreover, stress can be linked to several key factors, such as the presence or lack thereof of conspecifics, opportunities to hide, and exposure to potential predators. In captivity, animal care staff, visitors, and researchers may be perceived as predators. Moreover, the cleaning of their cages should be considered to be stressful. Not only does animal care staff enter the habitat, the removal of scent markings may also cause subsequent social stress. As scent marking is an important aspect of the natural behavior of marmosets, also in laboratory settings [52], it is preferred to limit the removal of scents as much as possible. Other causes of chronic stress in captivity should be the topic of further research [16]. Regarding cleaning, it was suggested that chromosomal disorders in marmosets could be related to the chemical disinfection of their environment [53].

Additionally, in laboratory settings, animals are often kept as monogamous pairs, which differs significantly from what is observed in the wild. In the wild, dominant females have helpers to assist them in raising their offspring, also known as alloparenteral carrying [54]. The absence of social structures could be stressful and may be associated with the occurrence of MWS in captivity. More research is needed to elucidate the role of specific social structures in the occurrence of MWS.

### 4.4. Other Factors

It has been proposed that urinary lipid metabolites may be involved in the development of MWS [55]. These metabolites are produced from polyunsaturated fatty acids and are excreted in urine. They regulate a wide range of inflammatory responses. Lipid metabolite levels in urine from seven healthy and seven MWS-affected marmosets were measured. Forty-one types of metabolites were detected in the urine samples of both groups. Among them, arachidonic acid-derived metabolites accounted for 63% (26/41 types) of all detected metabolites. The levels of arachidonic acid-derived prostaglandin (PG) E2, PGF2α, thromboxane B2, and F2-isoprostanes were significantly increased in the urine samples of MWS-affected marmosets. The clinical implications of urinary lipid metabolites and their role in MWS need to be further investigated.

The histological examination of six marmosets suffering from non-diarrheic weight loss, diagnosed as MWS-affected marmosets, revealed the presence of adult flukes (*Platynosomum* sp.) in the liver [27]. These flukes were also observed inside the gallbladder, as well as in the intra- and extrahepatic bile ducts. It was suggested that *Platynosomum* sp. has a role in the etiology of MWS. The prevalence of *Platynosomum* sp. infection is reported to be 8.9% in free-ranging marmosets. The prevalence of *Platynosomum* sp. in captive callitrichids is currently unknown [56,57].

## 5. Diagnostic Methods

Based on consistent clinical signs (see Section 3) and a history of emaciation, greasy, thinning, or unkempt haircoat, decreased body weight, hind leg paralysis, muscle atrophy, alopecia, chronic diarrhea without infectious cause, a failure to thrive, and general debility, an assumptive diagnosis of MWS can be made [4,5,7,8,16,58].

Currently, a presumptive antemortem diagnosis is made based on consistent clinical signs and a history of progressive weight loss. However, weight loss is an ambiguous clinical sign of which its significance may be difficult to quantify. Body condition scores, on the other hand, provide a more objective assessment of an animal’s general condition. It was reported that a body weight < 325 g identified most marmosets affected by MWS, although some small or juvenile animals may also fall into this range. Moreover, progressive body weight loss of 0.05% per day from peak body weight was found in all MWS-affected marmosets, suggesting the possibility of making a presumptive diagnosis of MWS based on weight loss [15]. However, although sensitivity was optimal, no specificity was reported. Hypoproteinemia (serum albumin < 3.5 g/dL), anemia (hematocrit < 35%), elevated liver enzymes, and thrombocytosis are considered as indications of MWS [15,16,18,19].

It would be ideal to have reliable, noninvasive, objective tests for the diagnosis of MWS. As described earlier (see Section 4.1), elevated α1-P1 levels in feces indicated protein loss via the GIT resulting from intestinal damage. An enzyme-linked immunosorbent assay (ELISA) for measuring α1-P1 in marmosets was developed [38]. Fecal samples were collected from seven MWS-affected and eight healthy marmosets. Because of its similar molecular weight, α1-P1 was used as a proxy for albumin. Under similar conditions, α1-P1 is also able to pass into the lumen of the GIT. Additionally, α1-P1 is resistant to enzymatic digestion and bacterial degradation in the GIT, allowing fecal analysis of α1-P1. Although measurement of α1-P1 constitutes a potential means of diagnosing MWS, its sensitivity and specificity have not been evaluated. Although more research is needed, measurement of urinary lipid metabolites may also be a diagnostic tool [55]. Other reports proposed elevated levels of calprotectin and serum matrix metalloproteinase 9 (MMP9) as potential biomarkers for MWS [19,59]. MMP9 is thought to play a role in the pathogenesis of inflammatory conditions, like IBD in humans, which, like MWS, is characterized histologically by the presence of inflammatory cell infiltrations in the intestines. Seven MWS-affected and seven healthy marmosets were included in this study. MMP9 levels were significantly increased in marmosets with MWS compared to healthy animals. MMP9 is suggested as a useful biomarker for the diagnosis of MWS [19,59].

MWS has great similarities to IBD in humans. Therefore, the potential of using commercial human colored chromatographic immunoassay IBD rapid tests to detect MWS in marmosets was investigated. These rapid tests were able to demonstrate high fecal concentrations of calprotectin and lactoferrin in IBD patients [50]. Therefore, 77 fecal samples of 11 callitrichid species were tested involving 23 MWS-affected animals and 54 control animals. The absence of lactoferrin in all samples likely indicates that callitrichid and human lactoferrin do not cross-react with the mouse monoclonal antibodies used in the test. Calprotectin tested positively in almost 65% of marmoset cases, however, demonstrating low specificity [50]. Therefore, the test was judged unsuitable for the diagnosis of MWS.

Gluten is expected to be involved in MWS. As described earlier, the assessment of plasma IgA-AGA and IgA-CIC levels could indicate MWS [25,40].

The definitive diagnosis of MWS requires the evaluation of affected tissues—usually obtained postmortem (see Section 3)—in addition to noninvasive diagnostics. However, recently, a technique for the collection of minimally invasive endoscopic biopsies of the GI tract has been described for common marmosets [60]. This could be a valuable tool to facilitate antemortem diagnosis so that treatment regimens may be started in earlier stages of the syndrome.

## 6. Treatment Options

The high prevalence of MWS necessitates effective treatment. However, as the exact underlying etiology is unclear, developing an effective treatment method is difficult. Historically, recommended treatments have focused on antibiotics, anti-inflammatory therapeutics, and supportive care [4,8,61].

As infectious agents have been suggested as a possible etiology for at least some of the clinical signs, affected animals often exhibit initial improvement with antibiotic and antiparasitic therapy [4,61]. In addition to the antibacterial action, tetracyclines have been shown to enhance GI mucosal protection in humans, and hypothetically also in callitrichids [62,63,64].

As discussed earlier, serum MMP9 plays a role in the pathogenesis of MWS and could be useful for the diagnosis of MWS [19]. Treatment with an MMP9-targeted inhibitor showed promising results in rats and shows promise as a specific treatment for MWS. Nevertheless, more research is required before definitive claims can be made.

Supportive care includes providing a high-protein diet, administering lactobacilli, fluid therapy, and improving diet palatability [4].

During treatment, clinical signs such as diarrhea and dehydration usually decrease in severity and even might resolve completely [61]. The degree of weight loss typically slows or stabilizes, and weight gain may even be observed [8]. It is possible to manage afflicted animals on these supportive treatments for a significant amount of time, but ultimately, the prognosis remains poor since the underlying cause of the syndrome is not addressed. Relapse inevitably occurs if supportive care is withheld.

### 6.1. Anti-Inflammatory Treatment

The use of anti-inflammatory medications shows good results, likely due to a direct effect on GI inflammation [8,61].

In humans, glucocorticoids are widely employed as a basic treatment for IBDs. In fact, treatment with glucocorticoids, orally for eight weeks, is one of the few therapies that also demonstrated an improvement in clinical signs of MWS in marmosets (*n* = 11) [58]. However, prednisolone, one of the most widely used glucocorticoids in humans, is associated with osteopenia [65]. As MWS-affected marmosets are assumed to be concurrently afflicted with metabolic bone disease and corresponding loss of bone density, long-term glucocorticoid therapy may not be desirable in laboratory-housed callitrichids but could be applied to individual cases [3,15,35,58,66]. Budesonide may be preferred over prednisolone owing to its high hepatic clearance, ability to reach high intestinal concentrations, and lower systemic concentration, subsequently leading to fewer adverse effects [67]. Long-term oral budesonide administration to MWS-affected marmosets improved body weights and increased serum albumin levels significantly over time [58]. Budesonide treatment can be initiated at 0.5 mg per animal orally once per day for eight weeks. The dose may be increased to 0.75 mg if the animal does not show an adequate response to treatment within eight weeks. The number of relapses was reported to be low. This study demonstrated that budesonide is a promising therapeutic option for marmosets chronically, but not acutely, affected by MWS. Another known immunosuppressant, azathioprine, commonly used in other species with IBD, could be beneficial in callitrichids with MWS [68,69].

### 6.2. Tranexamic Acid

Intraperitoneal administration of tranexamic acid, a plasmin inhibitor, with supportive measures is reported to be an effective treatment for MWS (*n* = 6) [18]. No notable adverse effects were observed. In addition, intestinal protein loss in MWS-affected marmosets was observed to be significantly reduced with this treatment [38]. Although this treatment protocol was effective, it was taxing on both animals and caregivers, and therefore a modified protocol was developed [70]. Here, the administration route of tranexamic acid was subcutaneous. The tranexamic acid solution was not diluted in this modified protocol to reduce contamination risk. The frequency of administration of the iron formulation and tranexamic acid solution was reduced in the modified method to reduce load on the animals as well as the caregivers. The modified method consisted of a subcutaneous injection of undiluted tranexamic acid five times per week, 2.0 mL of amino acid formulation for intravenous injection, 0.1 mL of vitamin formulation for subcutaneous injection, and oral administration of iron formulation five times per week. There were significant treatment effects in the modified method, such as increased hematocrit, serum albumin, and body weight. In addition, treatment effects on the animals’ appearance were also observed. Pretreatment emaciation, arched back, rough fur, and tail alopecia were observed, whereas no abnormal appearance was noted after treatment. The average treatment period was 37.8 ± 25.34 days, which was shorter than that of the original method (56 days) [70].

### 6.3. Dietary Changes

Studies reporting on the marmoset microbiome are increasing in number. We know that the gut microbiome is intimately connected to diet and gastrointestinal health. The captive marmoset microbiome is dominated by *Bacteroidetes*, *Bifidobacterium*, *Proteobacteria*, *Fusobacteria*, and *Actinobacteria* [71,72,73,74,75]. The GIT of captive marmosets is enriched with *Enterobacteriaceae*, while that of wild marmosets is enriched with *Bifidobacterium* [73]. *Bifidobacterium* is believed to process host-indigestible carbohydrates such as tree gum and gum Arabic. To that point, it has been suggested that *Bifidobacterium* sp. could potentially serve as a probiotic dietary supplement for laboratory-housed marmosets.

A change in diet from a biscuit to gel diet proved to be beneficial to the health of callitrichids [76]. The biscuit diet was a commercially processed New World primate biscuit that contained 20% animal-based protein. The gel diet involved a commercially processed callitrichid gel, containing 20% insect-based protein. Both diets were supplemented with a variety of small treats. Marmosets fed primarily the biscuit-based diet exhibited a significant decline in body weight. Both male and female marmosets exhibited a substantial improvement in clinical health presentations, including body weight, after transition to the gel-based diet. In addition, a significant improvement in reproductive success post-diet change was reported. Interestingly, *Bifidobacterium* sp. was increased after transition to the gel-based diet [34,76].

Unfortunately, not many controlled studies on the application of probiotics in NHPs have been performed yet. Probiotics are expected to help reduce the incidence of diarrhea in captive NHPs by supporting the natural microflora in the GI tract. Although the scientific background is not presented, SD Pro Mini tablets for marmosets are available online (https://www.bio-serv.com/product/SDPROMINI.html (accessed on 8 November 2024)). Administration is unlikely to cause harm and may benefit the microbiome [77]. Nevertheless, more research is required to investigate the role of probiotics in the prevention of MWS in callitrichids.

## 7. Prevention and Control

MWS is only observed in captivity. Therefore, we should take the natural biological and ethological needs of these animals into account. Marmosets are sentient beings with complex behaviors, and they require enriched, multidimensional environments, a varied diet, and the opportunity to fulfill their natural biological and ethological needs. Enclosures should contain bark to gouge and for scent marking, which are both important ethological needs for marmosets. Marmosets require a familiar environment, full of familiar scents. Therefore, it is preferrable to limit the frequency of cleaning of their enclosures and to not use chemical disinfectants [1,53]. Their enclosures should not be isolated from auditory communication, as marmosets have a complex vocal communication system and can use vocalizations to calm down conspecifics. Conversely, they can also signal problems or dangers through specific alarm calls, such as when a researcher, visitor, or caretaker approaches. In zoos, when selecting species in nearby exhibits, special consideration should be given to sounds or smells that may be perceived as predators of callitrichids [16]. Their enclosures should be equipped with several hiding places and sleep boxes.

Marmosets are unable to synthesize vitamin D_3_ without access to UVB radiation. Therefore, marmosets maintained indoors require dietary provision of vitamin D_3_ due to lack of sunlight exposure. Most commercial diets for callitrichids contain relatively high levels of vitamin D_3_ [2,8,35]. The survey of the EAZA requested information on whether the enclosure had access to natural sunlight [16]. No conclusion could be drawn from their results concerning the impact of direct sunlight access or not on MWS prevalence. Preferably, captive callitrichids should have the ability to expose themselves to direct sunlight. If not, full-spectrum fluorescent bulbs providing UVB radiation should be placed close to the cages [78].

Animals should be fed a nutritious and balanced diet formulated for their species. The overall protective effect of restricting concentrated feed and soluble carbohydrates and increasing fiber concentration should be considered to help prevent the development of MWS [16,33]. Additionally, the beneficial effect of a gluten-free diet on marmosets with MWS has been reported [25].

Close monitoring of food and water intake, as well as fecal and urine output, is mandatory. Monthly weigh-ins are important for the early detection of MWS in the colony. By carefully monitoring subtle trends in body weight and condition, and by monitoring blood parameters such as albumin and hematocrit, the early detection of callitrichids with MWS may become commonplace. Early diagnosis is crucial, and likely results in higher treatment success rates, better stabilization, or even reversal of the disease process. Treatments (see Section 6) should be initiated as soon as possible to halt progression.

Parasitic, viral, and bacterial infections should be treated, as marmosets are particularly susceptible to viral and bacterial infections due to their limited inter-individual variation in both MHC Class I and II loci [32].

By means of pedigree analyses, a monogenetic hereditary defect was excluded in a captive colony of MWS-affected marmosets in 2005 [23]. The authors assumed that MWS is a multifactorial disease with exogenous and endogenous contributing factors. Currently, genetics are assumed to play a role in the etiology of IBD in humans, and hypothetically also in the etiology of MWS. In humans, over 200 genetic loci are associated with IBD [79,80,81,82,83,84]. In marmosets, IBD-associated enteritis was also shown to upregulate pro-inflammatory immune responses in the duodenum and jejunum [72]. Further studies are needed to elucidate the immunological mechanisms involved and to elucidate how and which genes contribute to the pathogenesis of MWS. Therefore, although not evidence-based, genetic profiling and the removal of animals exhibiting signs of MWS from the breeding pool may be considered.

## 8. Discussion

One of the most devastating noninfectious GI diseases in callitrichids is MWS. This syndrome often presents with weight loss, muscle atrophy, diarrhea, anemia, hypoproteinemia, and enteritis, among other signs, or with acute moribundity or death. Determining treatment options is difficult, because the exact etiology of MWS remains unclear. Often, a multimodal treatment plan is implemented to address multiple potential risk factors. Clearly, further research is instrumental to the development of population management plans targeted at reducing the incidence of MWS. To this end, advances in diagnostic and treatment modalities need to be researched.

There are multiple diseases that could also cause progressive weight loss and a deteriorating general body condition in callitrichids, such as amyloidosis and IBD. Recently, novel GI disease presentations with relatively high morbidity and mortality in marmoset colonies were reported in Japan and the USA [85,86]. In Japan, reported clinical signs were vomiting, bloating, and weight loss concurrent with significant dilation of the descending duodenum [85]. In the USA, reported clinical signs were diarrhea, weight loss, or poor weight gain, abnormal hematology, and serum biochemistry values concurrent with duodenal ulcers/strictures [86]. The etiology of these novel GI disease presentations is yet to be confirmed, but the clinical signs resemble MWS.

The sample size in several studies was relatively small [18,19,27,38,50,58,68]. Furthermore, analytical validation is required when applying immunoassays for the first time in new species. Moreover, it is difficult to interpret survey results from different institutions [2,9,16]. To that end, such results require cautious interpretation and should be used as indications for the establishment of novel research goals.

## 9. Conclusions

MWS is a unique and challenging problem in callitrichids in captivity. MWS is clinically characterized by untreatable progressive weight loss concurrent with numerous other clinical signs like muscle atrophy, tail alopecia, chronic diarrhea, anemia, and hypoproteinemia. Although morbidity and mortality related to MWS is high in captive callitrichids, its specific cause is still unclear. Early identification and diagnosis by recently developed noninvasive urine and fecal tests of affected animals show promise, and continued investigation is of greatest importance. Promising treatments have been developed that involve glucocorticoids, tranexamic acid administration, probiotics, and diet optimization. More research is warranted to fully understand the underlying etiology and to further refine diagnostic and therapeutic methods for MWS in captive callitrichids. For now, preventative measures should be aimed at stress reduction and optimizing animal housing, husbandry, and nutrition.

## Figures and Tables

**Figure 1 vetsci-12-00203-f001:**
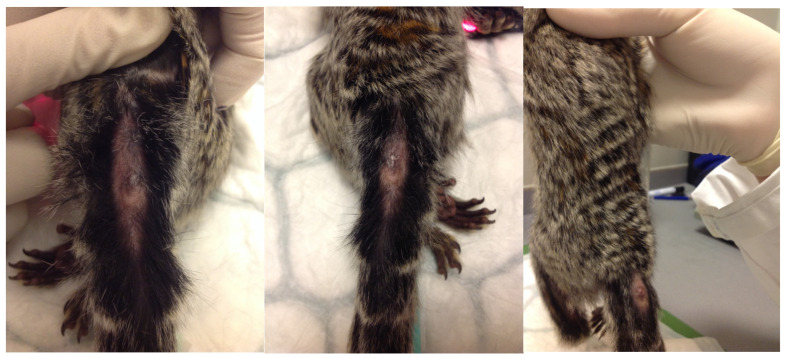
Three examples of alopecia, typically located at the base of the tail in Marmoset Wasting Syndrome, affected common marmosets housed at the Biomedical Primate Research Centre (BPRC), Rijswijk, The Netherlands (photographs provided by BPRC).

## Data Availability

Not applicable.

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
