# Peer review of "Recent Advances in the Etiology, Diagnosis, and Treatment of Marmoset Wasting Syndrome"

_vetsci, 2025, doi:10.3390/vetsci12030203_

Round 1
Reviewer 1 Report
Comments and Suggestions for Authors
The paper is well-written and describes the Marmoset Wasting Syndrome overall. The manuscript includes important information on marmoset care.
Author Response
28 January 2025
Re: Manuscript ID vetsci-3369933
Dear reviewer 1,
Please find a revised version of our review manuscript ‘Recent Advances in the Etiology, Diagnosis, and Treatment of Marmoset Wasting Syndrome’ for reconsideration for publication in Veterinary Sciences. In preparing this revised version of our manuscript, we would like to thank you for your helpful comments in order to improve its overall quality. To this end, we have considered all of your suggestions.
Below are our responses to your comments.
Comment - The paper is well-written and describes the Marmoset Wasting Syndrome overall. The manuscript includes important information on marmoset care.
Reply – Thank you for your positive reply.
If there are any additional concerns and/or comments about the revised manuscript, please do not hesitate to contact me.
We look forward to receiving your final decision in the very near future.
Yours sincerely,
Jaco Bakker

Reviewer 2 Report
Comments and Suggestions for Authors
This paper is a review of Marmoset Wasting Syndrome (MWS), a disorder that affects only captive marmosets. The paper discusses the suspected causes of MWS, reviews its diagnosis and treatment, and offers recommendations for the future based on recent veterinary findings.
Overall, the article is clear and well written in all its parts; each paragraph develops the different topics clearly and in depth but I have some suggestions and important comments but I have some suggestions and important comments.
I would like to preface this by stating that I am not a veterinarian, but a primatologist, so I do not have the skill to review the medical aspects (specific treatments etc.) considered in the paper. However, I accepted this review because I have worked with marmosets in the wild, as well as with a captive colony in a laboratory and in a sanctuary where rescued marmosets and tamarins are rehabilitated.
These are my comments:
1. Please specify the keywords used to conduct the review in the Methods section.
2. In my opinion, this paper lacks important considerations regarding the ecological and ethological needs of marmosets. It primarily presents medical findings without considering other crucial factors that may contribute to the development of MWS.
Below, I provide some insights into these additional factors.
First, there is no mention of the living conditions of marmosets that contracted MWS. The paper does not examine the captive environments in which they live or compare the incidence of MWS under different conditions. For example, it would be useful to analyse whether cases of MWS occur in zoos and sanctuaries and how these settings might influence its prevalence.
On the living condition we find just these sentences: Line: 357: “The provision of hiding areas and nestboxes in the enclosures can reduce the stress level. In zoos, when selecting species in nearby exhibits, special consideration should be given to sounds or smells associated with perceived predators of callitrichids [1,16].”
Living conditions probably have a significant impact on the occurrence of MWS. As the authors describe, MWS is likely a multifactorial syndrome in which diet and stress play a crucial role. For marmosets, stress is linked to several key factors, such as the presence of conspecifics, opportunities to hide, exposure to potential predators (in laboratory settings, researchers may be perceived as predators—personal observation), and the ability to engage in natural behaviors such as tree gouging, hole-making, and scent-marking.
Additionally, in laboratory settings, family groups typically consist of only a monogamous pair, which differs significantly from what is observed in the wild, where dominant females have helpers to assist them in raising twins. This social structure should be considered when discussing the potential causes and prevention of MWS.
Moreover, sunlight is one of the factors that can influence bone loss (one of the symptoms of MWS, as the authors report), and it is also linked to the immune system.
I wonder why the authors do not consider this crucial aspect. Is there any research on this?
All laboratories should have the ability to expose marmosets to sunlight.
I have worked in a sanctuary where MWS has never been detected in 30 years. There, marmosets live in a highly enriched environment and are exposed to natural sunlight. They can gouge tree bark whenever they want and mark trees. They live in social groups but do not reproduce, as it is a sanctuary.
Marmosets need a familiar environment, full of familiar scents. Their enclosures should not be isolated from auditory communication. Marmosets have a complex vocal communication system and can calm each other with simple vocalisations. Conversely, they can also signal problems or dangers through specific alarm calls—such as when a researcher approaches (personal observation).
In the laboratory where I worked, the marmosets lived under significant stress: confined to small cages with no access to sunlight, frequently cleaned cages that removed their scent markings (which could be highly stressful), no places to hide, and no bark to gouge. Despite many precautions, mortality rates were high.
Marmosets need to gouge trees—they are obligate exudate feeders and likely ingest small amounts of bark while doing so, benefiting from the nutrients it contains. They have a specialised microbiota capable of digesting plant cells and a complex gut. Studies have shown that tree gums and exudates they consume are rich in antibacterial and antifungal compounds. Additionally, they eat hundreds of different fruit varieties, including their seeds.
In conclusion, research that aims to understand (and resolve) a complex syndrome found only in captivity, without taking into account the animals' natural biological and ecological needs, is not a proper approach. Marmosets are sentient beings with complex lives, and they require a enriched - multidimensional - environments, a varied diet, and the opportunity to fulfill their natural biological and ecological needs.
All these factors should be taken into account when conducting a critical review, rather than just the strictly medical aspects. The limitations and challenges of captive conditions should be highlighted, especially when seeking solutions.
Furthermore, as a primatologist, I strongly advocate for a significant reduction—leading to the eventual elimination—of the use of these sentient beings in laboratory experiments, as many alternative methods are now available.
Author Response
5 february 2025
Re: Manuscript ID vetsci-3369933
Dear reviewer 2,
Please find a revised version of our review manuscript ‘Recent Advances in the Etiology, Diagnosis, and Treatment of Marmoset Wasting Syndrome’ for reconsideration for publication in Veterinary Sciences. In preparing this revised version of our manuscript, we would like to thank you for your helpful comments in order to improve its overall quality. To this end, we have considered all of your suggestions.
Below are our responses to your comments.
Comment -
- Please specify the keywords used to conduct the review in the Methods section.
- In my opinion, this paper lacks important considerations regarding the ecological and ethological needs of marmosets. It primarily presents medical findings without considering other crucial factors that may contribute to the development of MWS.
Below, I provide some insights into these additional factors.
First, there is no mention of the living conditions of marmosets that contracted MWS. The paper does not examine the captive environments in which they live or compare the incidence of MWS under different conditions. For example, it would be useful to analyse whether cases of MWS occur in zoos and sanctuaries and how these settings might influence its prevalence.
On the living condition we find just these sentences: Line: 357: “The provision of hiding areas and nestboxes in the enclosures can reduce the stress level. In zoos, when selecting species in nearby exhibits, special consideration should be given to sounds or smells associated with perceived predators of callitrichids [1,16].”
Living conditions probably have a significant impact on the occurrence of MWS. As the authors describe, MWS is likely a multifactorial syndrome in which diet and stress play a crucial role. For marmosets, stress is linked to several key factors, such as the presence of conspecifics, opportunities to hide, exposure to potential predators (in laboratory settings, researchers may be perceived as predators—personal observation), and the ability to engage in natural behaviors such as tree gouging, hole-making, and scent-marking.
Additionally, in laboratory settings, family groups typically consist of only a monogamous pair, which differs significantly from what is observed in the wild, where dominant females have helpers to assist them in raising twins. This social structure should be considered when discussing the potential causes and prevention of MWS.
Moreover, sunlight is one of the factors that can influence bone loss (one of the symptoms of MWS, as the authors report), and it is also linked to the immune system.
I wonder why the authors do not consider this crucial aspect. Is there any research on this?
All laboratories should have the ability to expose marmosets to sunlight.
I have worked in a sanctuary where MWS has never been detected in 30 years. There, marmosets live in a highly enriched environment and are exposed to natural sunlight. They can gouge tree bark whenever they want and mark trees. They live in social groups but do not reproduce, as it is a sanctuary.
Marmosets need a familiar environment, full of familiar scents. Their enclosures should not be isolated from auditory communication. Marmosets have a complex vocal communication system and can calm each other with simple vocalisations. Conversely, they can also signal problems or dangers through specific alarm calls—such as when a researcher approaches (personal observation).
In the laboratory where I worked, the marmosets lived under significant stress: confined to small cages with no access to sunlight, frequently cleaned cages that removed their scent markings (which could be highly stressful), no places to hide, and no bark to gouge. Despite many precautions, mortality rates were high.
Marmosets need to gouge trees—they are obligate exudate feeders and likely ingest small amounts of bark while doing so, benefiting from the nutrients it contains. They have a specialised microbiota capable of digesting plant cells and a complex gut. Studies have shown that tree gums and exudates they consume are rich in antibacterial and antifungal compounds. Additionally, they eat hundreds of different fruit varieties, including their seeds.
In conclusion, research that aims to understand (and resolve) a complex syndrome found only in captivity, without taking into account the animals' natural biological and ecological needs, is not a proper approach. Marmosets are sentient beings with complex lives, and they require a enriched - multidimensional - environments, a varied diet, and the opportunity to fulfill their natural biological and ecological needs.
All these factors should be taken into account when conducting a critical review, rather than just the strictly medical aspects. The limitations and challenges of captive conditions should be highlighted, especially when seeking solutions.
Furthermore, as a primatologist, I strongly advocate for a significant reduction—leading to the eventual elimination—of the use of these sentient beings in laboratory experiments, as many alternative methods are now available.
Reply – Thank you for your feedback and interesting comments.
Although out of the scope of this manuscript, we fully agree that primates should only be used for biomedical research when absolutely no alternatives are available. The development of alternative methods should indeed result in to the elimination of the use of primates in laboratory experiments. Nevertheless, as long as primates are housed in zoos and research facilities, their husbandry and veterinary care should be optimized. Our review will hopefully contribute to better husbandry and housing conditions, and improve welfare for captive callitrichids.
Unfortunately, although your observations are correct, we cannot implement your personal communications without supporting data in our paper. At the BPRC, we also have the gut feeling that the unlimited breeding of callitrichids costs too much energy, which results in them subsequently developing MWS. Our retired animals may get to become 15-25 years old without showing signs of MWS, and our oldest marmoset couple was 26-yrs old! However, there is insufficient data to properly support and publish these gut feelings.
As requested, we have now added the keywords which we used to perform our literature review in the methods section of the introduction.
A survey performed in zoos by Cabana, et al., 2018 (reference 16) investigated the incidence of MWS under different housing conditions in zoos and sanctuaries, involving stress and sunlight exposure. In our manuscript, we refer several times to this survey. No further peer-reviewed literature is available on this topic. Therefore, in order to keep the manuscript evidence-based, we refrain from speculating too much. The specific risk factors are now further discussed in ‘4. Etiology, pathogenesis and predisposing factors’, but are also addressed in ‘7. Prevention and control’. Readers may associate these risk factors with the specific housing conditions in zoos and sanctuaries. Please note that we have now emphasized the importance of appropriate housing conditions, and highlighted key ways in which these conditions are often infringed upon in laboratories and zoological institutions. As such, we now discuss the importance of social structures, scent marking, lighting regimens, and (dietary) enrichment, among others. We have also elaborated on the roles of sunlight and Vitamin D.
Please note that, in order to highlight the importance of these factors, we have now explicitly addressed them in the conclusion and abstract.
If there are any additional concerns and/or comments about the revised manuscript, please do not hesitate to contact me.
We look forward to receiving your final decision in the very near future.
Yours sincerely,
Jaco Bakker
Reviewer 3 Report
Comments and Suggestions for Authors
I have now read the manuscript with the tittle of “Recent Advances in the Etiology, Diagnosis, and Treatment of Marmoset Wasting Syndrome, which was submitted in Veterinary Sciences.
The authors have reported as integrated data from secondary sources; however, the recent data in depth and the significant interest are still missing.
I disagree with several sections, and they are need to rewrite and add more data including Sections 4 (Etiology, pathogenesis, and predisposing factors), 5 (Diagnostic methods), and 6 (Treatment). By returning to the literature, and reading it more in depth, the authors will find that the findings are sure as a updated data with previous reports. The author should consider rewriting and/or breaking down each section into subtopics.
Specific Comments:
1. Please indicate the number of animals used in each study discussed in section 2 (Lines 66-75).
2. Replace the term "symptom" with "clinical sign," as "symptom" refers to the indications reported by the patient to the physician.
3. Relocate the sentences discussing the ELISA test from lines 154-160 to the diagnostic methods section.
4. Include the references utilized for the descriptions in lines 261-269.
5. Revise the two paragraphs from lines 294-320 as they are quite difficult to follow, and add the references used in lines 310-320.
Author Response
5 February 2025
Re: Manuscript ID vetsci-3369933
Dear reviewer 3,
Please find a revised version of our review manuscript ‘Recent Advances in the Etiology, Diagnosis, and Treatment of Marmoset Wasting Syndrome’ for reconsideration for publication in Veterinary Sciences. In preparing this revised version of our manuscript, we would like to thank you for your helpful comments in order to improve its overall quality. To this end, we have considered all of your suggestions.
Below are our responses to your comments.
Comment - I disagree with several sections, and they are need to rewrite and add more data including Sections 4 (Etiology, pathogenesis, and predisposing factors), 5 (Diagnostic methods), and 6 (Treatment). By returning to the literature, and reading it more in depth, the authors will find that the findings are sure as a updated data with previous reports. The author should consider rewriting and/or breaking down each section into subtopics.
Reply – Thank you for your comment. You will find that large parts of the manuscript have been rewritten. We aim to discuss the topics pertaining to MWS that are currently most relevant. As a result, outdated studies or hypotheses have been largely omitted such as the parasite Trichospirura leptostoma and lack of vitamin E and selenium. The presented literature delineates the most important, recent developments surrounding MWS, and are therefore of most utility to veterinarians and animal management staff who aim to reduce the morbidity and mortality associated with MWS. Moreover, you will find that the manuscript now places more emphasis on animal husbandry and living conditions. This aligns with the multifactorial nature of MWS, and simultaneously aims to stimulate animal care staff to improve husbandry conditions for captive callitrichids. We also provide directions for future research, based on the most recent developments. Lastly, we choose to use paragraphs to delineate concepts and topics. For vastly different topics, we have now also decided to implement subheaders (e.g., see chapters 4.1. – 4.5. and 6.1. – 6.3.).
Specific Comments:
Comment 1. Please indicate the number of animals used in each study discussed in section 2 (Lines 66-75).
Reply – The number of animals used are indicated in the revised version of our manuscript.
Comment 2. Replace the term "symptom" with "clinical sign," as "symptom" refers to the indications reported by the patient to the physician.
Reply - We fully agree and replaced ‘symptoms’ with ‘clinical signs’.
Comment 3. Relocate the sentences discussing the ELISA test from lines 154-160 to the diagnostic methods section.
Reply - We agree that this information should be under the diagnosis subheader. Moreover, the number of animals used for developing this ELISA test has been added.
Comment 4. Include the references utilized for the descriptions in lines 261-269.
Reply – References are added as requested.
Comment 5. Revise the two paragraphs from lines 294-320 as they are quite difficult to follow, and add the references used in lines 310-320.
Reply – References are added as requested and both paragraphs are seriously revised to improve readability.
If there are any additional concerns and/or comments about the revised manuscript, please do not hesitate to contact me.
We look forward to receiving your final decision in the very near future.
Yours sincerely,
Jaco Bakker
Reviewer 4 Report
Comments and Suggestions for Authors
This paper is a nice review of currently available literature. There is an extensive review of potential pathophysiologies and associated potential diagnostic capabilities. I would recommend that the authors include additional information within the pathology section such as histology photos. Additionally, at the end of the paper the author's mention genetics. I would recommend significant more information about the genetic profiles and methods to prevent wasting disease using breeding management.
Author Response
28 January 2025
Re: Manuscript ID vetsci-3369933
Dear reviewer 2,
Please find a revised version of our review manuscript ‘Recent Advances in the Etiology, Diagnosis, and Treatment of Marmoset Wasting Syndrome’ for reconsideration for publication in Veterinary Sciences. In preparing this revised version of our manuscript, we would like to thank you for your helpful comments in order to improve its overall quality. To this end, we have considered all of your suggestions.
Below are our responses to your comments.
Comment - This paper is a nice review of currently available literature. There is an extensive review of potential pathophysiologies and associated potential diagnostic capabilities. I would recommend that the authors include additional information within the pathology section such as histology photos. Additionally, at the end of the paper the author's mention genetics. I would recommend significant more information about the genetic profiles and methods to prevent wasting disease using breeding management.
Reply – Many thanks for your recommendations to include additional information in the pathology and genetic section. We have carefully considered your suggestions, and have decided to omit the inclusion of histology photos, as findings can be highly diverse and aspecific. The inclusion of such histological images would not add to the overall quality of the manuscript, as no MWS-specific changes can be identified. This, in contract to the inclusion of Figure 1, depicting MWS-typical lesions. In the pathology section we added the use of immunohistochemistry. We added ‘Phenotypic characterization of the intestinal inflammation should be completed using immunohistochemical techniques, which reveals a T-cell mediated local immune response [22,23].’
We also elaborated the genetic profile part and breeding management paragraph to ‘By means of pedigree analyses, a monogenetic hereditary defect was excluded in a captive colony of MWS affected marmosets in 2005 [23]. The authors assumed that MWS is a multifactorial disease with exogenous and endogenous contributing factors. Currently, genetics are assumed to play a role in the etiology of IBD in humans, and hypothetically also in the etiology of MWS. In humans, over 200 genetic loci are associated with IBD [73-78]. In marmosets, IBD-associated enteritis was also shown to upregulate pro-inflammatory immune responses in the duodenum and jejunum [67]. Further studies are needed to elucidate the immunological mechanisms involved, and to elucidate how and which genes contribute to the pathogenesis of MWS. Therefore, although not evidence-based, genetic profiling and removal of animals exhibiting signs of MWS from the breeding pool may be considered.’
If there are any additional concerns and/or comments about the revised manuscript, please do not hesitate to contact me.
We look forward to receiving your final decision in the very near future.
Yours sincerely,
Jaco Bakker

Round 2
Reviewer 2 Report
Comments and Suggestions for Authors
The work has been revised in some parts according to the given suggestions and now incorporates several behavioral aspects that were initially omitted. I consider it ready for publication.
Reviewer 3 Report
Comments and Suggestions for Authors
All are clear and could you please re-check the journal and reference patterns.
Best,